



# 1 Impacts of sewage irrigation on soil properties of farmland in China: A review

Qiangkun Li[1]   Jiao Tang[2]   Tian Wang[3]   Dafu Wu[2*]   Carlos Alberto Busso[4]   Ruifeng Jiao[2]   Xiujuan Ren[2]
[1]Yellow River Institute of Hydraulic Research, Yellow River Conservancy Commission, Zhengzhou, 450003, China;
[2]School of Resource and Environment, Henan Institute of Science and Technology, Xinxiang, 453003, China:
[3]Third Institute of Geo-exploration Institute, Henan Bureau of Geo-Exploration& Mineral Development, Zhengzhou, 451464,
China;
[4]Departamento de Agronomía-CERZOS (CONICET: Consejo Nacional de Investigaciones Científicas y Tecnológicas de la
República Argentina), Universidad Nacional del Sur, San Andrés 800, 8000 Bahía Blanca, Argentina;
Correspondence to: Dafu Wu (uau9393@163.com)
Abstract:Fresh water is a valuable nonrenewable resource and plays an important role of maintaining
economic and social development. Sewage irrigation was taken a supplementary way to relieve water
resource shortage in some areas (such as North China). With extensive implementation of sewage
irrigation, serious pollution and destruction for farmland have gradually become a heated issue,
impeding basic industries sustainable development in the near future. Here in this paper, some soil
physical properties (soil bulk density, soil resistance to penetration and field capacity) and chemical
properties (pH, soil organic matter, nitrogen, phosphorous, potassium, heavy metal and organic
pollutants contents) and biological characteristics (soil microorganisms and enzyme activities) of
farmland affected by sewage irrigation were systematically reviewed on the base of the current
utilization status of China and some valuable suggestions were put forward to the future development
prospects. This review will be beneficial for promoting healthy development of sewage irrigation and
providing theoretical support for reclamation and high efficiency of effluents in China.
**Keywords:** wastewater irrigation, soil characteristics, agriculture, pollution, China



## Introduction

Water is not only a valuable natural resource, which maintains people's survival and development, but also constitutes the main constituent elements of environment (Bouwer, 1994; Gu et al., 2017; Molles, 2008; Piao et al., 2010). China is rich in total water resources volume, where the total amount of fresh water resources could reach $2.81*10^{12}$ m$^3$ that accounts for about 6% of global water resources (Thomas, 2008; Zhang and Wang, 2007). However, the per-capita water resources volume ($2.3*10^3$ m$^3$) of China is quite limited, which equals to 25 percent of the world average level, making China to be one of the poorest countries or regions (Fatta-Kassinos et al., 2011; JR, 1991; Wang et al., 2008). Even more serious is that the water resource characteristics of regional and seasonal distribution in China, to some extent, hinder the sustainable development of economic and social development in water shortage areas (Piao et al., 2010).

As a vast agricultural country, China has 70 percent of the total water being consumed in agricultural production or irrigation (Qadir et al., 2010; Shi et al., 2014; Yang, 2000). At present, about 50 percent of total cultivated land areas could be irrigated, which produce about 75 percent of nation grain output and more than 80 percent of cotton as well as more than 90 percent of vegetables in total (Jin and Young, 2001a; Liu and Xu, 2002; Zeng and Zhu, 2004). With the rapid development of national economy and continuous improvement of people's quality of life, water demand from its swelling urban and industrial sectors exacerbates water resources scarcity, these shortfalls can be filled only by diverting water from agriculture(Bouwer, 1994; Brown and Halweil, 1998; Vörösmarty et al., 2000). In China, there is just about $3.0*10^{10}$ m$^3$ of water in agricultural water shortage per year, resulting in a reduction in grain yield of $2.5-4.0*10^{10}$ kg (Brown, 1995; Wang et al., 2010; Zhou, 2002). At the same time, the total amount of wastewater discharged from industrial and urban areas is increasing, and the discharge is relatively concentrated, which is not affected by seasonal change and flooding. Most of the wastewater untreated or without necessary pretreatment is directly poured into rivers, lakes and



reservoirs, posing a potential threat to the environmental protection (Qadir et al., 2010; Rusan et al.,

53 2007).

Water resources depletion in agriculture promotes a large amount of sewage being used for
irrigation directly on a global scale, nearly $2.0 * 10^6 \, km^2$ involved 50 countries are irrigated by sewage
(Abaidoo et al., 2010; Qadir et al., 2007; Wallach et al., 2005). By the end of 2009, China has $5.9 *10^5$
$km^2$ irrigated farmland, accounting for 7.3 percent of the total area of irrigation farmland (Fang, 2011).
On the same time, sewage irrigation is using more at regions such as northern China where water
resources deficit is more serious. Application of sewage becomes to be an important supplementary way
to relieve agricultural irrigation water shortage water (Qadir et al., 2010; Rusan et al., 2007). Previous
studies reported that the main sewage irrigation area are Haihe river basin, Liaohe river basin, Yellow
river basin and Huaihe river basin, occupying approximate 85 percent of the sewage irrigation areas
(Liu and Xu, 2002).
Techniques of sewage treatment and reutilization have achieved the dual purpose of water
conservation and pollution control for some developed countries (Angelakis et al., 1999; Fatta-Kassinos
et al., 2011; Wallach et al., 2005). For China, sewage treatment techniques lag behind and most of water
quality for irrigation has not reached the standards for a long time, on the other hand, the sewage
irrigation management and monitoring system are not sound (Fang, 2011). More subjects were paid
attention to whether long-time sewage irrigation has affected soil properties of farmland in China (Khan
et al., 2008; Liu et al., 2005; Meng et al., 2016; Tang et al., 2004).
This paper systematically reviewed the effects of sewage irrigation on soil physical, chemical and
biological characteristics of farmland in China based on the development and utilization of China,
putting forward suggestions to the development prospects in the near future. The specific objectives
were to promote the sustainable development of sewage irrigation in China and provide theoretical
support for reclamation and high efficiency of effluents.



## 1. History of sewage irrigation

In fact, sewage irrigation does not use production and domestic sewage directly, but it is an engineering measure adopting appropriate treatment effluent which meets irrigation quality requirement for farmland, grassland landscape and groundwater recharge. It makes full use of soil self-purification ability purposefully and solves the lack of water resources and achieves sewage reclamation eventually (Liu and Xu, 2002; Qadir et al., 2010; Xia and Wang, 2001). Many developed countries in the world realized early on the strategic significance of sewage reutilization. Several western European countries used sewage irrigation since the middle of 17 century. Berlin, Germany is the oldest sewage irrigation sites in the world where approximate 100 km$^2$ of marginal and low-productivity land has been sewage irrigated since 1870 (Hass et al., 2012; Lottermoser, 2012). The first large-scale utilization of sewage irrigation country is America, where a suit of water purification system was assembled in 1920 and some research and intensive utilization of sewage irrigation were conducted (Chen et al., 2000; Sabol et al., 1987). Up to now, its wastewater treatment technology and application scope keep a leading place in the world. As one of the world's most severely water-deficient countries, Israel has established a comprehensive sewage system and sewage treatment projects in all its cities and settlements (Chen and Zhou, 2001; Heukelekian, 1957; Jueschke et al., 2008). Almost all of the wastewater could be effectively processed and utilized (Wallach et al., 2005). More than 57 percent of sewage after purification has been used in agriculture, garden and lawn irrigation, which accounts for about 20 percent of the total irrigation water, becoming a paragons of water resources efficient utilization countries (Chen and Zhou, 2001). Other countries, such as Tunisia, India, Jordan and Mexico, have also carried out relevant researches on wastewater irrigation and already accumulated a wealth of experience(Abu-Sharar et al., 2010; Bouri et al., 2008; Siebe and Fischer, 1996; Singh et al., 2012).

For sewage irrigation safety, different countries and international organizations have created a series of standards in practice (EPA, 1992; FAO, 1985). In 1973, the World Health Organization (WHO) published health guidelines for wastewater reclamation of farmland irrigation and aquaculture, which



claimed that the significance of sewage irrigation for farmland. Some indexes in the guidelines were
reset and new guidelines published in 1989. However, when those standards were popularized
throughout the world, few countries and regions adopted because of overly strict criteria (EPA, 1992).
The Food and Agriculture Organization (FAO) has also issued two technical reports about wastewater
treatment and irrigation reclamation and sewage quality controlling based on the current sewage
irrigation utilization in the worldwide (Pescod, 1992; Wescott, 1997). The water quality requirements
and sewage treatment methods for agricultural irrigation were also discussed and some guidance of
sewage irrigation was proposed in view of the actual situation of the national development level.
**2.   Application of sewage irrigation in China**

111        Compared with some developed countries, the source of sewage irrigation in Cnina mainly comes

from untreated or raw domestic and industrial wastewater (Pedrero et al., 2010). Though there has been
a long history for peasants using human wastes to fertilize farmland in many parts of China, sewage
irrigation development emerged later given the development level of economic and urbanization (Liu et
al., 2005; Zeng et al., 2007; Zhang, 2014). Three periods could be divided in accordance with the
development scale and stage: the first period is classified as spontaneous irrigation stage using sewage
effluents (Liu and Xu, 2002). Peasants lived in the suburban of Beijing began to mix domestic and
industrial effluents for farmland irrigation in the 1940s. But the emission load of sewage is relative
limited on a small scale, the national sewage irrigation area was just only $1.16 *10^2 \, km^2$ (Li and Luo,
1995; Zhang, 2014). The second period is served as preliminary development stage from 1957 to 1972.
In 1957, the Chinese government constructed sewage irrigation projects and the ministry of construction
engineering, agriculture and health jointly processed sewage irrigation into national scientific research
projects, prompting its preliminary development and forming a certain scale. And the first pilot scheme
for sanitary management of sewage irrigation was promulgated four years later (Zhang, 2014). Stepping
into 1970s, especially the implementation of reformation and opening policy and household contract
responsibility system accelerated rural enterprise development, sewage irrigation entered a



fast-developing period and had to face with unprecedented historical challenges. Firstly, some problems

of water resources shortage were gradually emerging and sewage irrigation areas increase dramatically.

More than $3.62 *10^4$ km$^2$ of farmland in China was irrigated using sewage effluents at the end of the

20th century (Wang and Lin, 2003). Although the Chinese government brought out and revised a series

of irrigation water quality standards which were applied to surface water, groundwater, aquaculture

treated wastewater and farmland irrigation water from agricultural production effluents in 1979, 1985

and 1992, respectively. Some standards for organic pollutants controlling were also increased and these

standards became national mandatory standards (Shi et al., 2008). However, just like many laws and

regulations in China, these standards existed in name only (Fang, 2011; Jin and Young, 2001b). And the

industrial and domestic sewage quality changed dramatically, toxic and harmful organic pollutant

species increased continuously (Weber et al., 2006). Only current water quality indicator standards

could not meet to the requirement of sewage irrigation, has the Chinese government come to realize the

hazards of sewage irrigation for agricultural production (Liu et al., 2005; Shi et al., 2006). A file of work

arrangement based on soil environmental protection and comprehensive adjustment in later period was

finally reported by General Office of the State Council in 2013. That was the first time for authority to

express explicitly prohibition of using wastewater containing heavy metals, refractory organic pollutants

and sludge, dredging of sediment, tailings that were untested or safety disposal for agricultural

production. Unfortunately, the relevant standards or guidelines of wastewater irrigation applied to new

conditions have been slow to appear due to various economic benefits.

## 3. Influence of sewage irrigation on soil properties

Soil is not merely the base supporting plant growth and breeding, but also the foundation of human

agricultural production (Killham, 1994; Molles, 2008). All kinds of human agricultural production

activities conduct in soil and abundant agricultural products acquire directly or indirectly from soil. Soil

is located in the interface of atmosphere, lithosphere, hydrosphere, and biosphere, participating a variety





of processes involved physics, chemistry, biochemistry and becoming the crucial place of material
circulation and energy exchange (Huang, 2000; Killham, 1994). Its existence provides a relatively stable
survival and procreation environment for aboveground vegetation and underground microorganism (Li
et al., 2000).

156       In China, peasants always use untreated sewage for farmland irrigation in agricultural production

directly. For substances dissolved in sewage, there are mainly four approaches of transference after
migrating into the soil (Wang and Lin, 2003; Zeng and Zhu, 2004). Some would gradually be reduced
by soil self-purification; some would be adsorbed and retained in the soil layer; some could be absorbed
by crops and the rest would enter aquifers along with water infiltration (Keesstra et al., 2012; Qadir et
al., 2010). Although soil has a certain capability of clearance and degradation of pollutants via
metabolism and transformation, increasing some nutritive element contents and enriching soil,
long-term irrigation by sewage which does not conform water quality standards induces organic
pollutants, heavy metals, solid suspended particles and bacteria microbes into the soil easily
(Fatta-Kassinos et al., 2011; Meng et al., 2016; Rusan et al., 2007; Zeng et al., 2007). Furthermore, the
worse thing is that these contents have been far beyond the ability of soil self-purification, causing
serious soil pollution and giving rise to some changes of soil physical, chemical and biological
characteristics.

## 3.1 Effect on soil physical characteristics

171       Long-term sewage irrigation upsets the balance of nature, causing ecological deterioration of

farmland (Wiel-Shafran et al., 2006). For the effects of sewage irrigation on soil physical properties, the
most direct performance is structural damage, functional disturbance and soil hardening (Wang and Lin,
2003). Soil bulk density is one of the important indicators measuring soil physical properties. It reflects
the degree of compaction to a certain extent, which has a great influence on soil aeration, soil



water-holding quantity and absorption capacity, infiltration, soil erosion resistance ability and solute
migration (Huang, 2000).The porosity of soil is subjected to changes in soil density (Neves et al., 2003).
A study on the consecutive irrigation in calcareous soil of China showed that long-term sewage
irrigation changed soil structure significantly (Li, 2001). Soil porosity and bulk density had a close
correlation with sewage irrigation time. As time goes on, the soil porosity decreased while the bulk
density increased. Furthermore, irrigation by effluents containing high salinity made soil secondary
salinization easily and enhanced total alkalinity and sodium alkalinity sharply, causing soil hardening
and soil permeability decrease (Li et al., 2003; Li et al., 2006). There are also reported that some organic
matter, microorganism, fibers and sediments from sewage deposited in the soil surface exerted an
negative impacts on soil physical traits, which would result in soil permeability degradation and soil
compaction occurrence (Siebe and Fischer, 1996).
The most conspicuous phenomenon of soil hardening is soil resistance to penetration, which is an
important index measuring crop roots elongation resistance (Barber, 1994; Mullins et al., 1994).
Generally, it is related to soil aggregate characteristics and soil particular spatial arrangement (Ahmed et
al., 1987; Franzluebbers et al., 2000). A study in Weihe River irrigation area by Hu found that the
topsoil of farmland within 10 cm irrigated by sewage directly was loose and resistance to penetration
was less than 500 kPa in this layer, which did not affect the crop root growth (Hu, 2010). However, the
value of resistance to penetration could be raised drastically with increase in soil depth, ranging from
415 kPa in 10cm to 1473 kPa in 45cm (Hu, 2010). Higher compaction in the deep soil layer shed a light
that there existed a thinning trend of unconsolidated for topsoil layer, which would compress thickness
of soil layer that was suitable for crop roots growth and increased the sensitivity of crop to
environmental changing. Problems like soil compactness cannot be neglected in future agricultural
production (Hern ández et al., 2015).
Field capacity refers to the maximum amount of water maintained by soil without being affected by
the groundwater and becomes the upper limit of available moisture for vegetation (Daniel, 1980; Qin,



2003). It is controlled by soil structure and soil texture, playing a vital role in farmland water balance controlling, irrigation and drainage, drought and moisture conservation (Shao et al., 2006). A research reported that the field capacity of loam would be greater than that of sand in normal conditions (Jia and Fan, 2007). When irrigated by sewage, the organic matter would be into the soil and increase the soil particles viscosity, thus increasing field capacity (Lan et al., 2010). Some similar research has confirmed that irrigation adopting eutrophic or untreated aquaculture wastewater did increase soil particles viscosity and enhanced field capacity (He, 2012; Wang and Lin, 2003).

### 3.2  Impacts on soil chemical characteristics

The effects of sewage irrigation on soil chemical properties are reflected in soil acidity-alkalinity firstly, which is one of the important factors affecting soil fertility (Bao, 2000). The formation and change of soil acidity-alkalinity depends on the relative strength of base substances leaching and accumulation process (Dheri et al., 2007). And the degree of acidity or alkalinity can be most conveniently expressed by pH value (Huang, 2000). Soil has a certain buffering function, thus the pH level is relatively stable (Masto et al., 2009). Once the value varies drastically, the soil chemical properties would be changed accordingly, which affects existing form, transformation and availability of soil nutrients directly (Ma and Zhao, 2010). Soil pH changes are related to irrigation water types and soil category (Wan et al., 2015). He et al.(2012) showed that the value of pH in soil would decrease with the increase of irrigation times when using wastewater from pig farms to irrigate the yellow clay. While an opposite conclusion was drawn that the value would increase if irrigated by effluents from paper-making factories to moderately degraded saline-alkali soil (Xia et al., 2011). It is also found that there was no obvious effect on vegetable field pH when sewage came from livestock breeding (Zhang et al., 2011). The reason for fluctuation of pH could be explained by the different degrees of ammoniation and nitrification of soil organic matter, anaerobic decomposition of organic matter, enrichment and release of metal ions (Dheri et al., 2007; Rusan et al., 2007).



Organic matter is a significant component of soil and its content was usually be regarded as an
important index measuring soil fertility (Rattan et al., 2005). The accumulation of soil organic matter is
not only closed related to natural environment conditions, but also depends on the input of organic
matter by all means (Qin, 2003; Shao et al., 2006). Sewage irrigation could solve water shortage in
actual agricultural production and increase soil fertility as well, which is the comprehensive reflection
of water and fertilizer. But the amplitude of increase showed the significant of differentiation in
different regions (Xue, 2012). A research irrigated by eutrophic sewage displayed that soil organic
matter contents of sandy soil and loam increased significantly, the value from loam increased by 97.1%
from 2.73 g/kg to 5.38 g/kg, while the value from sandy soil increased by 36.5% from 0.85 g/kg to 1.16
g/kg (Lan et al., 2010). Comparable differences also existed the degree of improvement in soil organic
matter with different soil layers. Extremely significant effects on soil organic matter were easily
discovered within 20 cm of topsoil, while the increase extent was significantly reduced with the
deepening of soil layers (Hu, 2010). Furthermore, more efforts in maintaining global carbon balance
have focus on the soil organic matter, which are considered as an irreplaceable role of affecting the
global warming in the worldwide. The accumulative effect of soil organic matter from sewage irrigation
has become one of inputs of soil organic carbon in farmland and participated in the global carbon
circulation (Rattan et al., 2005; Zhang et al., 2008b).
Nitrogen is an essential nutrient for crop growth, and the abundance and supply of nitrogen of soil
affects crop growth and development (Masto et al., 2009). A study of irrigation using aquaculture
wastewater for a long time displayed that nitrogen accumulation in soil increased significantly and the
nitrogen content in soil for more than 12 years' irrigation was significantly greater than that of untreated
farmland (Zhang et al., 2011). There existed an obvious feature of eutrophication when farmland was
irrigated by sewage over a long period of time, alkali-hydrolysable nitrogen content increased
significantly in each soil layers, especially in the soil surface, its content could reach the level of 8.26
mg kg$^{-1}$, much higher than the average (Hu, 2010). In the meantime, the nitrogen accumulation of soil

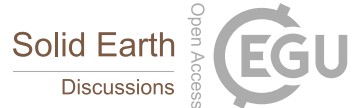

profile was significantly affected by nitrogen mobility and irrigative infiltration (Wiel-Shafran et al.,

2006). The accumulation of $NO_2^-$ and $NO_3^-$ downward migration easily with water eluviation when

polluted water was adopted for irrigation, causing groundwater pollution in a shallow layer of water

(Zhang et al., 1996). Sewage irrigation had a lesser impact on $NH_4^+$ existing in deep soil and

groundwater, but for the $NO_3^-$ concentration, great influence was emerged for long-term sewage

irrigated soil, causing groundwater pollution for deep soil layer easily (Liu and Lu, 2002).

Phosphorus is one of the three essential nutrients for plants. Not only it constitutes the components

of many important compounds in plants, but also participates in various plant metabolic processes by all

means (Dalai, 1977; Marschner et al., 2007; Redding et al., 2002). The study on farmland and forest

land found that the total phosphorus had significantly increased in the soil surface and most of them

could be kept in the topsoil within 40 cm when using pig farms' wastewater irrigation for a long-term

(Hu et al., 2010; Hu et al., 2012). Comparable concentrations were found in the farmland with piggery

wastewater irrigation that the phosphorus mainly accumulated in the plowing layer (0-40cm) and

increased with the increase of irrigation time (Yu, 2009). Reddling et al (2005) discovered that the

available phosphorus and total phosphorus content were significantly higher than that of controlling

irrigated by piggery wastewater after anaerobic digestion and the phenomenon of excessive

accumulation of phosphorus appeared in the top soil layer within 5 cm.

Potassium is also a major nutrient in higher plants, which together with nitrogen and phosphorus are

known as three essential factors of plant nutrition. Among them, the available potassium refers to the

potassium what is easily absorbed by the plant and becomes the main diagnostic index of soil fertility

(Huang, 2000; Qin, 2003). After sewage irrigation, the soil available potassium content has increased

greatly, the foremost reason is that there contains a lot of nutrients in the sewage, making available

potassium enrichment in the soil surface (Hu, 2010; Masto et al., 2009). The content of total potassium

in soil could be also improved for sewage irrigation, previous studies reported that the application of

alcohol wastewater has significantly increased the soil total potassium content, improving soil fertility





(Xu, 2007).
In general, heavy metals from sewage effluents could be adsorbed by soil particles and most heavy
metal ions are concentrated in the soil, resulting in soil heavy metal pollution, which has become the
most serious problem for human health (Liu et al., 2005; Mapanda and Mangwayana, 2005; Wan et al.,
2015). According to the bulletin of soil pollution published by Chinese government in 2014, 39 of 55
surveyed areas irrigated sewage existed soil contamination. 26.4 percent of investigation soil points in
total exceeded the maximum permitted levels and the main pollutants were cadmium (Cd), arsenic (As)
and polycyclic aromatic hydrocarbons (PAHs). Some heavy metal types were same, but heavy metal
spatial distribution existed heterogeneity (Hu et al., 2006; Khan et al., 2008; Liu et al., 2005). A broad
distinction of vertical distribution of heavy metals could be discovered in soil profile and it had been
recognized for many years that heavy metals are mainly concentrated in the soil within 50cm and these
vertical distribution varied with soil texture (Cao, 2004). Heavy metal contents corresponded to the
lithology structure in soil vadose zone. The silt was not favorable for heavy metal accumulation, the
sandy soil took the second. The highest content was found in loam, becoming the main enrichment of
heavy metals in soil (Wang and Lin, 2003). Besides, the degree of heavy metals enrichment in soil was
also closely related to sewage irrigation time and concentration of heavy metal ions in sewage (Liu et al.,
2005; Rattan et al., 2005). Wang et al found that long-term excessive irrigation by exceeded standard
sewage would pose a threat to soil, copper (Cu), lead (Pb), zinc (Zn), cadmium, arsenic, mercury (Hg),
chromium (Cr) and other harmful substances had been serious exceeded the limits of soil capability.
Five heavy metals (Cd, Cr, Cu, Zn, Pb) increased gradually during sewage irrigation in the farming
areas of Beijing and Shenyang and some elements like Cd, Cu, Zn, and Pb were excessive accumulation
in soil (Sun et al., 2006; Wang et al., 2006). Ultimately, these heavy metals are dangerous to human
health through various food chains (Loska et al., 2004) .
Except for heavy metal pollution in soil, various degrees of organic pollutants were existed in some
sewage irrigation areas (Qadir et al., 2010). Organic pollutants, such as aromatic hydrocarbons, phenols,



organic chlorine are easily discovered in industrial wastewater (Tian et al., 1993). A research by collecting seven different depths of soil samples irrigated by wastewater in Taiyuan, Shanxi Province found that the constituent parts in sewage were diversified and extremely complicated, main pollutants were plasticizer, such as the phthalate esters, skatol, sterols, polycyclic aromatic hydrocarbons and so on. The most serious pollution came from polycyclic aromatic hydrocarbons in soil. These substances could penetrate into the groundwater and relevant studies have documented in the nearby shallow groundwater (Du et al., 2010). A survey involved Shenyang and Fushun sewage irrigation area, the China's largest oil wastewater irrigated area, showed that the accumulation of toxic substances irrigated by petrochemical industrial sewage was serious, among which the aromatic hydrocarbons were quite a proportion, and carcinogens benzene severely exceeded (Zhang et al., 2003). The sensory indicators of rice produced in this region are extremely poor with strong smell from oils and aromatic compounds (Fang, 2011).

### 3.3 Influence of soil microorganisms and enzyme activities

Soil microorganisms, as an important part of maintaining soil quality, participate in most of soil biological and biochemical activities and are sensitive to reflect some changes of soil quality health (Stenberg, 1999). The composition and activities of soil microbe are dynamic processes with environmental change and the number of microbial living cells is regarded as one of the most sensitive biological indicators (Li et al., 2005a). Sewage irrigation would cause micro-environmental change to some extent, having a great effect on soil microbial activities (Zhang et al., 2008b). Soil bacteria, fungi and actinomycetes are the three essential types of microorganisms that can be used for reflecting the total amount of soil microorganisms and playing a significant role of soil organic matter and inorganic materials transformation (Aleem et al., 2003). The number of bacteria and actinomycetes in the soil showed a descending trend after long-term sewage irrigation, while the number of fungi increased slowly (Ge et al., 2009). Similar results in Shenyang and Fushun sewage irrigation region were found





that sewage irrigation changed the contents of soil nutrient and PAHs and then had a direct effect on the microbial populations, among which total nitrogen had a high significant positive and significant positive correlation with bacteria, nitrogen-fixing bacteria and phosphorus bacteria, respectively (Zhang et al., 2007; Zhang et al., 2008a). In the meantime, adoption of sewage irrigation equipment also affected the number of soil microorganisms (Heidarpour et al., 2007). A series of studies by Oron et al. displayed that soil surface humidity affected the total number of soil bacteria under the circumstance of sewage irrigation. When subsurface or underground drip irrigation was adopted, the total number of bacteria by subsurface drip irrigation is much higher than that of the underground. The most likely explanation would appear to be that soil played a role of secondary filter in the process of sewage infiltration, reducing contact probability between sewage and aboveground vegetation part (Oron et al., 1999; Oron et al., 1995).

Soil enzymes are the catch-all term of active substances found in the soil, primarily coming from the soil microbes and plant root secretion and enzymes released by decomposition of animal and plant residues (Burns and Dick, 2002; Cao et al., 2003). Common enzymes mainly include oxidoreductases, hydrolytic enzymes, crack enzymes and transferase, all of which participate in and promote a large proportion of organic substances transformation and material circulation by various of soil ecological processes (Zhang et al., 2011). Some relevant research has produced evidence to suggest that irrigation by petroleum wastewater could stimulate aerobic heterotrophic bacteria and fungi growth in soil, and soil dehydrogenase, catalase, polyphenol oxidase activities showed a positive correlation with the total petroleum hydrocarbon content, while soil urease activities had a significant negative correlation with the total petroleum hydrocarbon content in soil (Li et al., 2005b). Other observations were found that soil enzyme activities had been dually influenced by soil nutrient and PAHs pollution when irrigated by petroleum processing wastewater for a long time. The soil organic carbon and total phosphorus contents showed a significant relation with dehydrogenase, polyphenol oxidase and urease activities respectively, the content of PAHs were significant positive correlated to dehydrogenase and urease activities

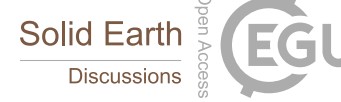

respectively, while it was significantly positively related to polyphenol oxidase activities (Zhang et al.,
2007; Zhang et al., 2008a). Similar studied of the relationship between soil heavy metal pollution by Pb
and Cd and the soil enzyme activities in a heavy industry city of Baoding, Hebei Province had also
proven that soil urease and hydrogen peroxide enzyme activities increased with the increase of Pb and
Cd in soil (Liu, 1996). There were problems with many studies concerning indirect influence caused by
sewage irrigation such that soil secondary salinization in calcareous drab soil, leading to enzyme
activities constrained, causing soil environment quality decline (Li, 2001).

## 4 Implications for sewage irrigation development in the future

For the current development of sewage irrigation in China, government officials must clearly realize
that wastewater discharge itself is a guarantee of water scarcity for grain production and huge
population demands for food security in China. On the other hand, we have to face the adverse effects
brought by the sewage irrigation and find solutions instead of complaints. However, any research and
development involve sewage irrigation need to consider the actual situation in China carefully. Four
proposals have been made for achieving safe and efficient utilization of wastewater irrigation in
farmland of China.

### 4.1 Strict control of pollution sources and reasonable management system

At present, the first question for sewage irrigation in China is to solve the quality problem gradually.
Starting from the sewage sources, water quality monitoring should be strengthened and water quality of
wastewater entering the farmland should be strictly controlled (Engineering, 2000; Qadir et al., 2010;
Yang, 2000). Excessive contaminated water is forbidden for discharging and utilization. In addition, in
the view of the current situation that governors always adopt an attitude towards removing supervision
responsibility and administration of sewage in practice, so scientific management system of sewage
irrigation should be established and implemented urgently, realizing explicit responsibility and
embodiment in different stages of sewage discharge, disposal and irrigation (Shi et al., 2014; Zeng and





Zhu, 2004). For companies and individuals in the wastewater irrigation areas, awareness of environmental protection should be increased. For those enterprises that have illegal discharges sewage behaviors, the amounts of punishment need to be much greater than its illegal profit. For individual, the health risks of sewage irrigation should be extensively published, enhancing awareness of environmental and human health (Rattan et al., 2005; Wan et al., 2015).

**4.2 Optimization mode of sewage irrigation and avoiding irrigation by single type sewage for a long time**

China has summarized some effective methods for sewage irrigation techniques through several decades of practice, which including oxidation pond purifying wastewater treatment and mixed irrigation between wastewater and clear water (Hong et al., 2011). However, the flood irrigation mode is the most common adopted in the vast area of sewage irrigation. It , on the one hand, wastes a lot of valuable sewage resources, and on the other hand causes serious soil pollution (Zhang et al., 2005). Therefore the current model of sewage irrigation should be changed and optimized, combining agricultural water-saving irrigation with sewage pretreatment and developing aerated drip sewage irrigation. Considering different crop growth stages, edible parts and contaminants in wastewater irrigation, sewage irrigation time and quantity need to be allocated reasonably, reducing the adverse effects of sewage irrigation on crop growth and human health (Oron et al., 1999; Oron et al., 1995).

**4.3 Conducting sewage irrigation adjusted measures to local conditions and protecting groundwater resources**

Because of unmatched irrigation facilities, improper irrigation methods, unscientific irrigation systems and low management level, there existed some problems that field irrigation efficiency was low and percolation towards deep soil layer was serious (Liu and Xu, 2002). Improper use of sewage irrigation equipment or irrigation by untreated sewage easily gave rise to pollutants infiltration in soil, endangering security of drinking water and even forming inverse funnel of sewage, which would pose a threat to deep unground water. Once groundwater is contaminated, it will be difficult to recover, and the



consequences will be severe. It seems reasonable to assume that it is not suitable for sewage irrigation for some inadaptable wastewater irrigation areas such as strong soil permeability, high underground water level, aquifer outcrop and centralized drinking water sources, which easily lead to groundwater pollution and be unfavorable to our human health (Qadir et al., 2010; Zhou, 2002). Therefore, developing corresponding scheme in line with local conditions was an appropriate measure to reduce the environmental risks caused by sewage irrigation.

**4.4 Adsorption and degradation of soil harmful substances by preferential absorption from plant and microbe**

In the remedy technology of heavy metal contaminated soil, phytoremediation is highly favored for its advantages such as excellent reinforced effect, low cost and high environmental benefit (Nie et al., 2016). At the same time, microorganisms could either fix heavy metal ions through their metabolic functions or convert toxic heavy metal ions into non-toxic or low-toxic prices (Li et al., 2015). Hyperaccumulators could be introduced to repair contaminated soil by adopting their own strong absorption abilities for some heavy metals and anti-heavy metal toxicity. Furthermore, the function of microbial selective absorption was jointly utilized to establish the system of bioremediation, improving the remediation efficiency of heavy metal pollution (Rajkumar et al., 2010).

**5. Conclusion**

The biggest challenge now facing the Chinese government is how to meet the soaring water demand of its expending urban and industrial portions without decreasing its own agriculture needs. In consideration of world food security, sewage irrigation on a mass scale indeed is an alternative way for water scarcity in some areas of China. However, the effects of inappropriate sewage irrigation on soil physical, chemical and biological characteristics of farmland should be paid attention to in practice. Some relevant matched irrigation facilities, proper irrigation methods, scientific irrigation systems from agriculture and water resources departments and high sewage management level together with clear reward and punishment from government are worthy of been put forward, popularized and applied in



line with local conditions, promoting reclamation and high efficiency of effluents in China.

## Acknowledgements

This study was financially supported by National Natural Science Foundation of China (51379085) and
Scientific Research Staring Foundation from Henan Institute of Science and Technology, China.

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
