# Peer review of "Impacts of sewage irrigation on soil properties of farmland in China: A review"

_Solid Earth, 2017_

## Referee Comment (RC1) · Anonymous Referee #1 · 13 Nov 2017

This paper aims to present a review of the literature on irrigation with sewage water in China. Unfortunately it does so mostly in a descriptive, qualitative mode. Most of the Chinese literature stems from grey sources or unknown journals with hardly any circulation. I suspect much of it is in Chinese, but the authors do not indicate if this is the case in the reference list. The discussion of the references is so superficial and incoherent that hardly any useful information is transferred to the reader, who is unable to access many of the cited sources if s/he is not based in China.

The purpose of the paper appears to be to convince the Chinese government to adopt much more stringent regulations to control irrigation with waste water. Apparently, untreated or poorly treated waste water from various sources has been used, including waste water from heavy industry and the petroleum industry. Unsurprisingly, this has

led to soil and groundwater pollution with hydrocarbons, heavy metals, and other com-
pounds.

Dramatic as this is, scientifically this is not new or surprising. Other countries have
developed wastewater irrigation practices much further and this progress is well doc-
umented in easily accessible journals. This paper adds no new knowledge to the
available literature on the subject. Furthermore, the heavy focus on China leads to
considerable bias in the quoted literature. Nationality of the authors is a poor criterion
to select which papers to include in a review, further limiting the value of the paper for
the international readership of SE.

In the annotated pdf I indicated which were poorly organized or otherwise unclear.
On a few occasions I corrected the English, but the text really should be completely
rewritten by an English editor.

Please also note the supplement to this comment:
https://www.solid-earth-discuss.net/se-2017-116/se-2017-116-RC1-supplement.pdf

[Figure]

**Supplement:**

[revised manuscript text omitted]

---

## Short Comment (SC1) · 17 Nov 2017

This manuscript reviewed the impacts of sewage irrigation on soil physicochemical properties and biological characteristics of farmland in China. The objectives of the review are clear. Sewage irrigation may cause organic and inorganic environmental pollution. The proposals in the manuscript have certain guiding significance in reasonable use of wastewater irrigation in China. However, the authors need to improve the manuscript, especially on the introduction part, as much useless information in this section. The authors should cite more English references in the paper. In addition, English should be polished by native speakers.

---

## Referee Comment (RC2) · Anonymous Referee #2 · 18 Nov 2017

Here is the content of the manuscript in a nutshell. 1. Sewage and other wastewater has been applied without proper treatment. 2. The observed consequences that are well known from the experience of other regions or countries. The Government must develop and enforce regulations with clear reward and punishment.

There is no specific information that the Government could use to develop the regulations.

the only novelty in the manuscript is the geographic region. In my opinion, there is no reason to publish this work on an international journal, since an international reader will not find any new concepts, technologies, or observations.

---

## Author Comment (AC3) · 28 Dec 2017

**Impacts of sewage irrigation on soil properties of a farmland in China: A review**

Qiangkun Li[1] Jiao Tang[2] Tian Wang[3] Dafu Wu[2*] Carlos Alberto Busso[4] Ruifeng Jiao[2] Xiujuan Ren[2]

[1]Yellow River Institute of Hydraulic Reseach,Yellow River Conservancy Commission, Zhengzhou, 450003, China;
[2]School of Resource and Environment, Henan Institute of Science and Technology, Xinxiang, 453003, China:
[3]Third Institute of Geo-exploration Institute, Henan Bureau of Geo-Exploration & Mineral Development, Zhengzhou, 451464, China;
[4]Departamento de Agronomía-CERZOS (CONICET: Consejo Nacional de Investigaciones Científicas y Técnicas de la República Argentina), Universidad Nacional del Sur, San Andrés 800, 8000 Bahía Blanca, Argentina;
Correspondence to: Dafu Wu (uau9393@163.com)

Abstract:Fresh water is a valuable nonrenewable resource and plays an important role in maintaining economic and social development. Considering its large population and consumption potential, water resources deficit will certainly impede basic industries a sustainable development in China in the near future. Application of sewage irrigation, to some extent, was regarded as an alternative way to solve the problem of agricultural irrigation water shortage in some areas (such as North China). However, accompanied with an extensive implementation of sewage irrigation, some problems on sewage irrigation in agriculture are gradually obvious, especially serious pollution and destruction of farmlands. In this paper, the effects of sewage irrigation on soil physical (soil bulk density, soil resistance to penetration and field capacity), chemical (pH, soil organic matter, nitrogen, phosphrous, patassium, heavy metal and organic pollutants) and biological characteristics (soil microorganism and enzyme activities) of farmlands in China were systematically reviewed on the base of the current utilization status of China's farmland sewage irrigation. Some feasible suggestions were put forward to the development prospects in the future. This review will be beneficial for promoting a healthy development of sewage irrigation and providing a theoretical support for reclamation and a high efficiency use of effluents in China.

**Keywords:** wastewater irrigation, soil characteristics, agriculture, pollution, China

**Introduction**

Water is not only a valuable natural resource that maintains people's survival and development, but also constitutes the main constituent element of the environment (Bouwer, 1994; Gu et al., 2017; Molles, 2008; Piao et al., 2010). China is rich in the volume of total water resources. In this country, the total amount of fresh water resources could reach $2.81*10^{12}$ m$^3$ which account for about 6% of the global water resources (Thomas, 2008; Zhang and Wang, 2007). However, the per-capita water resource volume is relatively limited (i.e., $2.3 *10^3$ m$^3$), considering the large population of China. It only represents 25 % of the world average level, becoming thus one of the poorest countries of water resources per capita (Fatta-Kassinos et al., 2011; JR, 1991; Wang et al., 2008). Even worse, the regional water resource characteristics and seasonal distribution in China constrain the sustainable development of economic and social development in water shortage areas (Piao et al., 2010).

As a vast agricultural country, extensive quantities of water are consumed in China's agricultural production. Irrigation water accounts for more than 70% of the total water consumption (Qadir et al., 2010; Shi et al., 2014; Yang, 2000). At present, there are about 50% of the total cultivated land that could be irrigated. This non-irrigated land currently produces about 75% of the nation grain output, more than 80% of cotton, and more than 90% percent of vegetables (Jin and Young, 2001a; Liu and Xu, 2002; Zeng and Zhu, 2004). With the rapid development of the national economy and the continuous improvement of people's quality of life, industrial and domestic water consumption increase continuously. This reduces the water available with agricultural purposes. In addition irrigation water is not guaranteed, and water shortage is becoming increasingly serious as result (Bouwer, 1994; Brown and Halweil, 1998; Vörösmarty et al., 2000). In China, agricultural water shortage per year is about 3.0 $* 10^{10}$ m$^3$, determining a reduction in grain yield of 2.5-4.0$*10^{10}$ kg (Brown, 1995; Wang et al., 2010; Zhou, 2002). At the same time, the total amount of water discharged as waste from industrial and urban areas is increasing. This waste water is relatively concentrated, and it is not affected neither by seasonal changes nor floodings. Most of the untreated waste water is poured directly into rivers, lakes and reservoirs, which determines a potential threat to the ecological environment (Qadir et al., 2010; Rusan et al., 2007).

Water resource depletion in agriculture results in that large amounts of sewage are used for irrigation on a global scale. Nearly $2.0*10^5 \, km^2$, involving 50 countries, are irrigated by sewage (Abaidoo et al., 2010; Qadir et al., 2007; Wallach et al., 2005). There were $4.3*10^{10} \, km^2$ of irrigated farmland worldwide by the end of 2009. This accounted for 7.3% of the total irrigated area in China (Fang, 2011). Sewage irrigation shows a rising trend, especially in northern China since it is the main area of water shortage. Application of sewage could solve the problem of water shortage for agricultural irrigation in this area (Qadir et al., 2010; Rusan et al., 2007). There are reports that sewage irrigation is currently focused on the Haihe, Liaohe, Yellow and Huaihe river basins, which represent about 85% of the area irrigated by sewage (Liu and Xu, 2002). In developed countries, the techniques of sewage treatment and reutilization have improve enough as to achieve the dual purposes of water conservation and pollution control (Angelakis et al., 1999; Fatta-Kassinos et al., 2011; Wallach et al., 2005). However, the sewage treatment techniques have lag behind in China and water quality will not reach good enough standards for a long time. Even worse, management of sewage irrigation and monitoring systems have been not appropriate (Fang, 2011). More attention was paid to determine it long-term sewage irrigation either affected or not the soil properties of farmlands in China (Khan et al., 2008; Liu et al., 2005; Meng et al., 2016; Tang et al., 2004).

This paper reviews the effects of sewage irrigation on soil physical, chemical and biological characteristics of farmlands in China, putting forward suggestions for the development of perspectives in the near future. This specific objective seek to (1) promote the sustainable development of sewage irrigation in China, and (2) provide a theoretical support for the high efficiency of sewage in reclamation programs.

**1. History of sewage irrigation**

Commonly, production is not the result of using domestic sewage directly. Engineering is needed to obtain effluents after applying appropriate treatments that will meet the irrigation quality requirements for (1) irrigating farmlands and grassland landscapes, and (2) contributing to the groundwater recharge, solving the lack of water resources and achieving eventually sewage reclamation (Liu and Xu, 2002; Qadir et al., 2010; Xia and Wang, 2001). Many developed countries in the world have realized quite early on the strategic significance of sewage reutilization. Western European countries began to use sewage to irrigate farmlands since the middle of the 16th century. Germany is considered to have the world's largest and oldest sewage irrigation sites. In this country approximately 100 km$^2$ of marginal and low-productivity land have been irrigated by sewage since 1928 (Hass et al., 2012; Lottermoser, 2012). The first country which made large-scale utilization of sewage irrigation was the U.S.A, where a suitable water purification system was assembled in 1920; some research and intensive utilization of sewage irrigation was conducted in this country (Chen et al., 2000; Sabol et al., 1987). To date, its wastewater treatment technology and application scope keep a leading place in the world. Japan has begun to recycling sewage and implemented rural sewage treatment projects since the 1960's. Approximately in the year 2000, small sewage treatment plants were implemented on a national scale, depending on small sewage treatment systems for agricultural irrigation in the 1970's (Francks, 1979; Morishita, 1988). As one of the most severely water-deficient countries in the world, Israel has established a comprehensive sewage system and sewage treatment projects in all its cities and settlements (Chen and Zhou, 2001; Heukelekian, 1957; Jueschke et al., 2008). Almost all of its wastewater was effectively processed and utilized (Wallach et al., 2005). More than 57% of the sewage after purification has been used for irrigation in agriculture, gardens and lawns. This accounts for about 20% of the total irrigation water. This made Israel a very efficient water utilization country (Chen and Zhou, 2001). Other countries, such as Tunisia, India, Jordan and Mexico, have also conducted relevant research on wastewater irrigation, and already accumulated a wealth of experience (Abu-Sharar et al., 2010; Bouri et al., 2008; Siebe and Fischer, 1996; Singh et al., 2012).

For sewage irrigation safety, different countries and international organizations have created a set of standards in practice (EPA, 1992; FAO, 1985). In 1973, the World Health Organization (WHO) published health guidelines of wastewater recycling for farmland irrigation and aquaculture. The guidelines claimed that sewage for farmland should be treated strictly. As a result, guidelines referring to some indexes were adjusted, and new guidelines were published in 1989. However, those standards were too rigid and of little practical value when they were made popular. The result was that most countries and regions did not follow them. The Food and Agriculture Organization (FAO) also issued two technical reports about wastewater treatment and irrigation recycling, and control of effluent quality controlling based on the current situation of sewage irrigation utilization worldwide (Pescod, 1992; Wescott, 1997). The water quality requirements and sewage treatment methods for agricultural irrigation were also discussed, and some guidance of sewage irrigation were proposed in view of the actual situation of the countries development levels,

.

**2. Application of sewage irrigation in China**

Compared with some developed countries, the source of sewage irrigation comes mainly from untreated or raw domestic and industrial wastewaters (Pedrero et al., 2010).There has been a long history on peasants in using human wastes to fertilize farmlands on many parts of China. The development of sewage irrigation emerged later given a level of economic development and urbanization for the country (Liu et al., 2005; Zeng et al., 2007; Zhang, 2014). There are three periods to take into account in accordance with the development scale and stage. The first period is classified as spontaneous irrigation because of using sewage effluents (Liu and Xu, 2002). Peasants who lived in the suburbans of Beijing began to mix domestic and industrial effluents for farmland irrigation in the 1940's. But considering that the emission loads of sewage were relatively limited to a small scale, the national sewage irrigation area was just only $1.16*10^2\,km^2$ (Li and Luo, 1995; Zhang, 2014). The second period is regarded as a preliminary stage of development from 1957 to 1972. In 1957, the Chinese government developed sewage irrigation projects. At this time, the Ministry of Construction Engineering, Agriculture and Health expanded sewage irrigation to national scientific research projects, prompting its preliminary development and forming a certain scale. The first pilot scheme for sanitary management of sewage irrigation was promulgated four years later (Zhang, 2014). Stepping into the 1970's, the implementation of reformation and opening policies, and household contract responsibility systems accelerated the development of urban and rural enterprises. Sewage irrigation entered a fast-developing period, and faced unprecedented historical challenges. Firstly, some problems of water resources shortage were gradually highlighted, and sewage irrigation areas increased dramatically as a result. More than $3.62*10^4\ km^2$ of farmlands in China were irrigated using sewage effluents at the end of the 20th century (Wang and Lin, 2003). Although the Chinese government brought out and revised a series of irrigation water quality standards, which were applied to surface water, groundwater, aquaculturetreated wastewater, and farmland irrigation, water came from effluents that were mainly agricultural products as raw material in 1979, 1985 and 1992. Some standards for controlling organic pollutants were also increased, and they became national mandatory standards (Shi et al., 2008). However, just like many laws and regulations in China, these standards only existed in name in practice (Fang, 2011; Jin and Young, 2001b). With the rapid development of the national economy, the industrial and domestic sewage water quality changed dramatically, and toxic and harmful organic pollutants increased continuously (Weber et al., 2006). Some current indicators of water quality standards could not adapt to the requirements for sewage irrigation. As a result the Chinese government came to realize about the hazards of sewage irrigation for agricultural production (Liu et al., 2005; Shi et al., 2006). A file of work arrangement on soil environmental protection and comprehensive adjustment for the near time was finally issued by the General Office of the State Council on January 28, 2013. This was the first time an authority explicitly prohibited the use of wastewater containing heavy metals, refractory organic pollutants and sludge, dredging of sediments, tailings that were untested, or safety disposal for agricultural production. However, the relevant standards or guidelines of wastewater irrigation including the new conditions have been delayed due to various economic benefits.

**3. Influence of sewage irrigation on soil properties**

Soil is not only the base for supporting plant both growth and breeding, but also the foundation of human agricultural production (Killham, 1994; Molles, 2008). All kinds of human agricultural production activities are mainly carried out in the soil, and abundant agricultural products are acquired directly or indirectly from it. Soil is located in the interface of the atmosphere, lithosphere, hydrosphere, and biosphere, participating in a variety of processes involving physics, chemistry and biochemistry, and becoming the crucial place of nutrient cycling and energy flux (Huang, 2000; Killham, 1994). Its existence provides a relatively stable survival and procreation environment for aboveground vegetation and underground microorganisms (Li et al., 2000).

In China, untreated sewage is often used directly for farmland irrigation in agricultural production. For substances dissolved in sewage, there are mainly four approaches of transference after migrating into the soil (Wang and Lin, 2003; Zeng and Zhu, 2004). Some of them would gradually be reduced by the soil self-purification; others would be adsorbed and retained in the soil layer; some of them could be
absorbed by the crops, and the rest would enter aquifers following water infiltration (Keesstra et al.,
2012; Qadir et al., 2010). Soil, to some extent, has the capacity to clear and degrade pollutants via
metabolism and transformation, increasing some nutrient content and fertility in the soil. However,
long-term irrigation using sewage that does not conform with water quality standards allows infiltration
of organic pollutants, heavy metals, solid suspended particles and bacteria microbes into the soil (Fatta-
Kassinos et al., 2011; Meng et al., 2016; Rusan et al., 2007; Zeng et al., 2007). These contents,
nevertheless, might be beyond the soil capacity of self-purification, causing serious soil pollution and
giving rise to some changes of soil physical, chemical and biological characteristics.

**3.1 Effects on soil physical characteristics**

Long-term sewage irrigation damages the balance of nature, causing ecological deterioration on
farmlands (Wiel-Shafran et al., 2006). Its most direct effect on soil physical properties includes
structural damage, functional disturbance and soil hardening (Wang and Lin, 2003). Soil bulk density is
one of the important indicators for measuring the effects on soil physical properties. It reflects the
degree of soil compaction to a certain extent, which has a great influence on soil aeration, soil water
holding and absorption capacities, infiltration, soil erosion resistance ability and solute migration
(Huang, 2000).The porosity of soil depends on changes in soil bulk density (Neves et al., 2003). A study
on the consecutive irrigation of calcareous soil in China showed that long-term sewage irrigation
changed soil structure significantly. Soil porosity and bulk density had a close correlation with sewage
irrigation time. As time increased from sewage irrigation, the soil porosity decreased while the soil bulk
density increased (Li, 2001). Furthermore, irrigation by effluents containing high salinity increased soil
salinization and enhanced both total and sodium alkalinity in the soil. This caused soil hardening and a
decrease in soil permeability (Li et al., 2003; Li et al., 2006). There are also reports that the organic
matter, microorganisms, fiber and sediments from sewage deposited in the soil surface exerted a
negative impact on soil physical traits, which resulted in a degradation of soil permeability and an
increased soil compaction (Siebe and Fischer, 1996).
The most conspicuous result of soil hardening is its resistance to penetration, which is an important index for measuring the resistance to crop root elongation (Barber, 1994; Mullins et al., 1994). Generally, it is related to soil aggregate characteristics and soil particular spatial arrangement (Ahmed et al., 1987; Franzluebbers et al., 2000). A study in the Weihe River irrigation area by Hu (2010) found that the topsoil of a farmland irrigated by sewage within the first 10 cm was directly loose, and its resistance to penetration was less than 500 kPa. As a result, it did not affect the crop root growth. However, its resistance to penetration became obvious deeper in the soil layer. This is because it was 415 kPa at 10cm soil depth and 1473 kPa at 45cm soil depth. Study of compactness determines that there exists a trend for an unconsolidated topsoil layer, which was suitable for crop root growth, and increased the sensitivity of the crop to environmental change. Problems like soil compactness cannot be neglected in future agricultural production (Hern ández et al., 2015).

Field capacity refers to the maximum amount of water maintained by the soil without including groundwater, and becomes the upper limit of available moisture for vegetation (Daniel, 1980; Qin, 2003). It is controlled by soil structure and soil texture, playing a vital role in controlling farmland water balance, irrigation and drainage, drought and moisture conservation (Shao et al., 2006). The field capacity of loam would be greater than that of sand under normal conditions (Jia and Fan, 2007). When irrigated by sewage, its organic matter would go into the soil and increase the soil particle viscosity, thus increasing field capacity (Lan et al., 2010). Some similar research has confirmed that irrigation with eutrophic or untreated aquaculture wastewater did increase soil particle viscosity and enhanced field capacity (He, 2012; Wang and Lin, 2003).

**3.2  Impacts on soil chemical characteristics**

Sewage irrigation can affect soil chemical properties through its effects on the soil pH. This is one of important factors which affect soil fertility (Bao, 2000). The formation and change of the soil acidity-alkalinity depends on the relative strength of the process of base substances leaching and accumulation (Dheri et al., 2007). The degree of acidity or alkalinity can be more conveniently expressed by the pH value (Huang, 2000). Since the soil has a certain buffer capacity, the pH value is relatively stable (Masto et al., 2009). Once the pH value varies drastically, the soil chemical properties will be changed accordingly, which affects directly the existing form, transformation and availability of soil nutrients (Ma and Zhao, 2010). Soil pH changes are related to the types of irrigation water and soil category
(Wan et al., 2015). He et al. (2012) showed that the soil pH value would decrease with the increase of
irrigation times using wastewater from hoggery to irrigate a yellow clay. On the other hand, Xia et al.
(2011) showed that the value of pH in soil increased if irrigated by effluents from paper-making
factories making a moderately degraded saline-alkali soil. It was also found that there was no obvious
effect on the soil pH of a vegetable field when sewage came from a livestock breeding enterprise
(Zhang et al., 2011). The reason for the fluctuation of the pH value could be explained by the different
degrees of ammonification and nitrification of the soil organic matter, anaerobic organic matter
decomposition, and release and enrichment of metal ions (Dheri et al., 2007; Rusan et al., 2007).

Organic matter is a significant component of the soil, and its content was usually be regarded as an
important index for measuring soil fertility (Rattan et al., 2005). The accumulation of soil organic
matter is not only closely related to natural environmental conditions, but it also depends on the input of
organic matter by all means (Qin, 2003; Shao et al., 2006). Sewage irrigation could solve water shortage
in the current agricultural production and also increases soil fertility. However, the amplitude of this
increase showed great agrotype and spatial differences (Xue, 2012). Irrigation by eutrophic sewage
showed that the soil organic matter content of sandy and loamy soils increased significantly. Values for
the loamy increased from 2.73 g/kg to 5.38 g/kg, (97.1%), while values for the sandy soil increased
from 0.85 g/kg to 1.16 g/kg (36.5%) (Lan et al., 2010). Comparable differences in the content of soil
organic matter also existed when considering increments of soil depth within the soil profile. Extremely
significant increases in the content of soil organic matter were easily discovered within the first 20 cm
of the topsoil, while the increase levels were significantly reduced with increases in soil depth (Hu,
2010). Furthermore, much effort in maintaining the global carbon balance have focused on the soil
organic matter, which is considered of having a unique role in affecting the global warming worldwide.
The cumulative effects of soil organic matter from sewage irrigation have become one of the inputs of
soil organic carbon in farmland and participated in the global carbon circulation (Rattan et al., 2005;
Zhang et al., 2008b).

Nitrogen is an essential nutrient for crop growth, and its abundance and supply in the soil affect crop
growth and development (Masto et al., 2009). A study of irrigation using aquaculture wastewater for a long time showed that nitrogen accumulation in soil increased significantly, and the nitrogen content in soil was significantly greater than that of the untreated farmland for more than 12 years (Zhang et al., 2011). There existed an obvious eutrophication when farmland was irrigated by sewage over a long period of time: alkali-hydrolysable nitrogen content increased significantly in each study soil layer, especially in the soil surface and its content could reach 8.26 mg kg$^{-1}$, much higher than the average (Hu, 2010). In the meantime, nitrogen accumulation in the soil profile was significantly affected by nitrogen mobility and irrigation infiltration (Wiel-Shafran et al., 2006). The accumulation of $NO_2^-$ and $NO_3^-$ because of water eluviation caused groundwater pollution at shallow layers when polluted kratos water was adopted for irrigation (Zhang et al., 1996). Sewage irrigation had a lower impact on $NH_4^+$ existing in deep soil and groundwater. However long-term sewage irrigated soil greatly influenced the $NO_3^-$ concentration, causing groundwater pollution from deep soil layer (Liu and Lu, 2002).

Phosphorus is one of the three essential nutrients for plants. Not only it constitutes the components of many important compounds in plants, but also participates in various metabolic processes in plants by all means (Dalai, 1977; Marschner et al., 2007; Redding et al., 2002). The studies on farmland and forest lands found that the total phosphorus had significantly increased in the surface soil and most of it could be kept in the upper soil (0-40 cm) using long-term wastewater irrigation (Hu et al., 2010; Hu et al., 2012). Comparable concentrations were found in farmlands irrigated with swine wastewater: phosphorus accumulated in the plowing layer (0-40cm) and increased with the advance of the time of irrigation (Yu, 2009). Reddling et al (2005) discovered that the soil available and total phosphorus contents were significantly higher than those from irrigation by piggery wastewater after anaerobic digestion, and phosphorus levels appeared to be a result of excessive accumulation in the top soil layer within 5 cm.

Potassium is also a major nutrient in higher plants; together with nitrogen and phosphorus they are known as the three essential factors for plant nutrition. Available potassium refers to the potassium that is easily absorbed by the plant and becomes the main diagnostic index of soil fertility (Huang, 2000; Qin, 2003). In studies of Hu (2010) and Masto et al. (2009), the soil available potassium content increased greatly after sewage irrigation. This was mainly because a lot of nutrients were contained in the sewage, making available potassium enrichment possible in the soil surface. The content of total potassium in soil could also be improved for the application of molasses alcohol water in the sewage irrigation; it significantly increased total soil potassium content, improving soil fertility (Xu, 2007).

In general, heavy metals from sewage effluents could be adsorbed by soil particles. Because of this most heavy metal ions are concentrated in the soil, resulting in soil heavy metal pollution. It has become the most serious problem for human health (Liu et al., 2005; Mapanda and Mangwayana, 2005; Wan et al., 2015). According to the bulletin of soil pollution published by the Chinese government in 2014, 39 out of 55 surveyed areas irrigated by sewage showed soil contamination by heavy metals. As much as 26.4% exceeded the maximum permitted levels of total of 1378 soil points, and the main pollutants were cadmium, arsenic and polycyclic aromatic hydrocarbons. This metal distribution was homogeneous, but heavy metal spatial distribution has shown important differences (Hu et al., 2006; Khan et al., 2008; Liu et al., 2005). A broad distinction of vertical distribution of heavy metal pollution was drawn in the soil profile. Cao (2014) concluded that heavy metal elements mainly concentrated in the soil within the first 50cm from the soil surface, and its vertical distribution varied with soil texture. Heavy metal content was related to the lithology structure in the soil vadose zone. Silt was not favorable for heavy metal accumulation and sandy soil took the second place. The main enrichment of heavy metals was reported in loamy soil (Wang and Lin, 2003). The degree of heavy metal enrichment in soil is also closely related to the timing of wastewater irrigation and the concentration of heavy metal ions in sewage (Liu et al., 2005; Rattan et al., 2005). Wang et al (2008) found that long-term, excessive irrigation by sewage exceeded the standards and posed a threat to soils. This is, Cu, Pb, Zn, Cd, As, Hg, Cr and other harmful substances seriously exceeded the limits of soil capacity. Five toxic metals (Cd, Cr, Cu, Zn, Pb) increased during sewage irrigation of farming areas in Beijing and Shenyang, and pollution with Cd, Cu, Zn, and Pb was exacerbated in soils (Sun et al., 2006; Wang et al., 2006). Ultimately, these heavy metals are dangerous to human health through various food chains (Loska et al., 2004).

In addition to heavy metal pollution in soil, there exist various other degrees of organic pollutants in some sewage irrigation areas (Qadir et al., 2010). Organic pollutants, such as aromatic hydrocarbons, phenols, and organic chlorines are easily discovered in industrial wastewaters (Tian et al., 1993). A research by collecting seven different soil samples depths irrigated by wastewater in Taiyuan, Shanxi Province, determined that the constituent parts in sewage diversified and were extremely complicated; they were pollutants as plasticizer, such as phthalate esters, skatole, sterols, polycyclic aromatic hydrocarbons and so on. The most serious pollution of polycyclic aromatic hydrocarbons was found in soil. This substance has penetrated into the groundwater and also been detected in the nearby shallow groundwater (Du et al., 2010). A survey in the Shenyang and Fushun sewage irrigation area, the China's largest oil wastewater irrigated area, showed that the accumulation of toxic substances because of the petrochemical industrial sewage was serious. Among the toxic substances, the aromatic hydrocarbons represented an important proportion, and the carcinogens benzene and pyrene were severely exceeded (Zhang et al., 2003). The sensory indicators of rice produced in this region were extremely poor, with strong smell from oils and aromatic compounds (Fang, 2011).

**3.3 Influence of soil microorganisms and enzyme activities**

Soil microorganisms, as an important part for maintaining soil quality, participates in most of soil biological and biochemical activities, and are sensitive to reflect changes of soil quality health (Stenberg, 1999). The quantity, composition and activity of soil microbial population are a dynamic process; environmental changes and the number of microbial living cells constitute one of the most sensitive biological indicators (Li et al., 2005a). Sewage irrigation would cause a change of microhabitat to some extent, having a great effect on soil microbial activities (Zhang et al., 2008b). Soil bacteria, fungi and actinomycetes can be used to reflect the total amount of soil microorganisms that play a significant role on soil organic matter and inorganic material transformation (Aleem et al., 2003). The number of bacteria and actinomycetes in the soil showed a descending trend after long-term sewage irrigation, while the number of fungi increased slowly (Ge et al., 2009). Similar results were found in the Shenyang and Fushun sewage irrigation region: sewage irrigation changed the content of soil nutrients and multiring hydrocarbon, and then had a direct effect on microbial populations. Their total nitrogen showed very significantly positive correlations with bacteria, nitrogen-fixing bacteria, and phosphorus bacteria (Zhang et al., 2007; Zhang et al., 2008a). In the meantime, the way of sewage irrigation application also affected the number of soil microorganisms (Heidarpour et al., 2007). A series of studies by Oron et al (1999) displayed that soil surface humidity affected the total number of soil bacteria after sewage irrigation. When subsurface or underground drip irrigation was adopted, the total number of bacteria of subsurface drip irrigation was much higher than that of the underground. The most likely explanation would appear to be that soil played a role of secondary filter in the process of sewage infiltration, reducing the contact probability between sewage and aboveground vegetation parts (Oron et al., 1999; Oron et al., 1995).

Soil enzymes are active substances found in the soil, primarily coming from the soil microbes and plant root secretions and enzymes released by the decomposition of animal and plant residues (Burns and Dick, 2002; Cao et al., 2003). Common enzymes mainly include oxidoreductases, hydrolytic enzymes, crack enzymes and transferation enzymes, all of which participate in and promote a large proportion of organic substance transformation and material circulation by various of soil ecological processes (Zhang et al., 2011). Some relevant research has produced evidence to suggest that irrigation by petroleum-processed wastewater could stimulate aerobic heterotrophic bacteria and fungi growth in the soil. The total petroleum hydrocarbon content in the soil showed a positive correlation with the soil dehydrogenase, catalase and polyphenol oxidase activities and a negative correlation with the soil urease activity (Li et al., 2005b). Other observations were found that soil enzyme activities were influenced by soil nutrients and multiring hydrocarbon pollution, after long-term irrigation by petroleum-processing wastewater. The soil both organic carbon and total phosphorus content showed significant relationships with the dehydrogenase, polyphenol oxidase and urease activities. The content of multiring hydrocarbon was significantly positive correlated with the dehydrogenase, urease, and polyphenol oxidase activities (Zhang et al., 2007; Zhang et al., 2008a). In Baoding, Hebei Province, a heavy industry city, it was determined that the soil urease and hydrogen peroxide enzyme activities increased with increases of soil Pb and Cd contents (Liu, 1996). Many studies have shown indirect influences caused by sewage irrigation such as soil secondary salinization in calcareous drab soil, which lead to constrain enzyme activities, causing a decline in soil environmental quality (Li, 2001).

**4.   Implications for sewage irrigation development in the future**

For the current development of sewage irrigation in China, governors must clearly realize that wastewater discharge itself is a guarantee to replace the water scarcity for grain production, and the huge population demands for food in China. On the other hand, the adverse effects brought by the sewage irrigation were removed completely. Therefore, any research and development involving sewage irrigation have to consider carefully the current situation in China. We suggest that four aspects should be taken into account for achieving a safe and efficient utilization of farmland wastewater irrigation in China.

**4.1 Strict control of the pollution and supervising systems**

At present, the first question for sewage irrigation in China is to solve the quality problem gradually. Starting from the sewage source, water quality monitoring should be much greater, and water quality of the wastewater entering the farmland should be strictly controlled (Engineering, 2000; Qadir et al., 2010; Yang, 2000). Contaminated water that seriously exceeds the threshold levels should be forbidden from discharging and utilization. Currently, governors always adopt an attitude towards removing responsibility of supervision and administration of sewage in the practice. Therefore, a management system of sewage irrigation should be established and implemented urgently. It has to explicit the appropriate responsibilities and embodiment at different stages of sewage discharge, disposal and irrigation (Shi et al., 2014; Zeng and Zhu, 2004). For the companies and individuals in the wastewater irrigation area, awareness of environmental protection should be increased. For that illegal discharge, sewage behavior enterprises punishment needs to be much greater than their illegal profit. For the individual, the health risks from sewage irrigation should be extensively published, enhancing awareness of environmental and human health (Rattan et al., 2005; Wan et al., 2015).

**4.2 Optimize the way of sewage irrigation, and avoid irrigation using a single-type-sewage for a long time**

China has summarized some effective methods for sewage irrigation techniques through several decades of practicing. They include oxidation pond purifying wastewater treatment, and mixed irrigation between wastewater and clear water (Hong et al., 2011). However, the flooding irrigation mode is the most common adopted in the vast area of sewage irrigation. On the one hand, it wastes a lot of valuable sewage resources. At the same time, however, it causes serious soil pollution (Zhang et al., 2005). Thus, the current model of sewage irrigation should be changed and optimized, combining agricultural water-saving irrigation with sewage pretreatment, and developing an underground, aerated drip sewage irrigation. Considering the different crop growth stages, edible parts and contaminants in wastewater irrigation, sewage irrigation time and quantity need to be allocated reasonably, reducing the effect of sewage irrigation on crop growth and human health (Oron et al., 1999; Oron et al., 1995).

**4.3 Control of sewage irrigation to local conditions and protecting groundwater resources**

Because of improper irrigation facilities and methods, unscientific irrigation systems and low management levels, field irrigation efficiency is low and percolation towards deep soil layers is serious (Liu and Xu, 2002). Failing in making a proper sewage irrigation or irrigation by untreated sewage easily gives rise to infiltration of pollutants in the soil, endangering the security of drinking water and even forming inverse funnels of sewage, which pose a threat to deep groundwater. Once groundwater is contaminated, it will be difficult to recover, and the consequences will be severe. It seems reasonable to assume that it will not be suitable for sewage irrigation for some unadaptable wastewater irrigation areas such as those with strong soil permeability, high underground water level, aquifer outcrops and centralized drinking water sources, which will easily lead to groundwater pollution and be unfavorable to human health (Qadir et al., 2010; Zhou, 2002). Therefore, it is appropriate to adjust measures to local conditions for sewage irrigation, and reduce the environmental risks caused by sewage irrigation.

**4.4 Adsorption and degradation of soil harmful substances by some plant and microbe characteristics of selective absorption**

In the technical field of heavy metal contaminated soil repair, phytoremediation is highly favored for its advantages such as an excellent reinforced effect, low cost and high environmental benefits (Nie et al., 2016). At the same time, microorganisms could either fix heavy metal ions through their metabolic functions or convert toxic heavy metal ions into non-toxic or low-toxic conditions (Li et al., 2015). Hyperaccumulators could be introduced to repair soils contaminated by sewage irrigation adopting their own strong absorption ability for some heavy metals and non-heavy metal toxicity. Furthermore, the function of microbial selective absorption was jointly utilized to establish the system of plant-microbial repairment, improving the efficiency of heavy metal pollution soil restoration (Rajkumar et al., 2010).

**5. Conclusion**

The biggest challenge that the Chinese government is now facing is how to meet the soaring water demand of its expanding urban and industrial portions without decreasing its own agricultural needs. In consideration of world food security, sewage irrigation on a mass scale is indeed as an alternative way for water shortage in some areas of China. However, the effects of inappropriate sewage irrigation on soil physical (soil bulk density, soil resistance to penetration and field capacity), chemical (pH, soil organic matter, nitrogen, phosphorus, potassium, heavy metal and organic pollutants) and biological characteristics (soil microorganisms and enzyme activities) in farmlands should paid attention to its practical application. Some relevant, proper irrigation facilities and methods, scientific irrigation systems from agriculture and water resource departments, and high sewage management levels are worth applying. This is together with clear rewards and punishments from the government, popularized and applied in line with local conditions, promoting reclamation and high treatment efficiency of effluents in China.

**Acknowledgements**

This study was supported by the National Natural Science Foundation of China (51379085) and the School of Resource and Environment, Henan Institute of Science and Technology, China.

[revised manuscript text omitted]

---

## Author Comment (AC4) · 28 Dec 2017

Thanks for your some valuable suggestions for the manuscript of sewage irrigation in China. Why we wrote this review of sewage irrigation in China farmland. Most of readers at abroad did not know applications of sewage irrigation in China currently. I have done some relative work for 30 years in China. The most potential place using sewage irrigation is China, we must understand the history and current situation and make better use of it in the near future. Now the Chinese government has adopted some regulations and laws, so some suggestions we gave are useful and exert some impacts on some developing counties facing water deficit.

Please also note the supplement to this comment:

https://www.solid-earth-discuss.net/se-2017-116/se-2017-116-AC4-supplement.pdf

---

## Editor Comment (EC1) · M. van der Ploeg (Editor) · 8 Jan 2018

The response of the authors to the comments of reviewer 1 was removed. The response did not comply with Solid Earth's general terms.